# Therapeutic Modulation of the Gut Microbiome by Supplementation with Probiotics (SCI Microbiome Mix) in Adults with Functional Bowel Disorders: A Randomized, Double-Blind, Placebo-Controlled Trial

**DOI:** 10.3390/microorganisms13102283

**Published:** 2025-09-30

**Authors:** Won Yeong Bang, Jin Seok Moon, Hayoung Kim, Han Bin Lee, Donggyu Kim, Minhye Shin, Young Hoon Jung, Jongbeom Shin, Jungwoo Yang

**Affiliations:** 1ILDONG Bioscience, Pyeongtaek-si 17957, Republic of Korea; yeong0417@ildong.com (W.Y.B.); moonjs@ildong.com (J.S.M.); young@ildong.com (H.K.); gksqls9131@ildong.com (H.B.L.); 2Department of Microbiology, College of Medicine, Inha University, Incheon 22212, Republic of Korea; ehdrb1548@inha.edu (D.K.); mhshin@inha.ac.kr (M.S.); 3School of Food Science and Biotechnology, Kyungpook National University, Daegu 41566, Republic of Korea; younghoonjung@knu.ac.kr; 4Division of Gastroenterology, Department of Internal Medicine, Inha University Hospital, Incheon 22212, Republic of Korea; 5Department of Microbiology, College of Medicine, Dongguk University, 123 Dongdae-ro, Gyeongju 38066, Republic of Korea

**Keywords:** functional bowel disorders, probiotics, clinical trial, human gut microbiome, serotonin, *Faecalibacterium prausnitzii*, *Blautia stercoris*

## Abstract

Functional bowel disorders (FBDs) are chronic gastrointestinal conditions characterized by recurrent symptoms associated with gut microbiota dysbiosis. Although accumulating evidence suggests that probiotics can improve symptoms in patients with FBD, the underlying mechanisms remain to be fully elucidated. In this randomized, double-blind, placebo-controlled clinical trial, 38 adults meeting the Rome IV diagnostic criteria of functional constipation (FC) and functional diarrhea (FD) received either a multi-strain probiotic complex or placebo for 8 weeks. Clinical outcomes were evaluated using the Irritable Bowel Syndrome Severity Scoring System (IBS-SSS), bowel habits questionnaire, and IBS Quality of Life (IBS-QoL) instrument. Fecal samples were collected at baseline and at week 8 for gut microbiota profiling via 16S rRNA gene sequencing and metabolomic analysis using gas chromatography–mass spectrometry. Probiotic supplementation significantly reduced the severity of abdominal bloating and its interference with quality of life, and improved the body image domain of the IBS-QoL. Beta diversity analysis showed significant temporal shifts in the probiotic group, while 16S rRNA sequencing revealed an increased relative abundance of *Faecalibacterium prausnitzii* and *Blautia stercoris*. Fecal metabolomic analysis further indicated elevated levels of metabolites implicated in the gut–brain axis. Multi-strain probiotic supplementation alleviated gastrointestinal symptoms and improved aspects of psychosocial well-being in adults with FBDs, potentially through modulation of the human gut microbiome.

## 1. Introduction

Functional bowel disorders (FBDs) are chronic gastrointestinal disorders with chronic functional symptoms, which are categorized into irritable bowel syndrome (IBS), functional constipation (FC), functional diarrhea (FD), functional abdominal bloating and distension and unspecified FBD [1]. Epidemiological studies estimate that FBDs affect approximately 11–20% of the global population, resulting in significant impairments of quality of life and increased healthcare utilization [2]. Patients usually present with recurrent abdominal pain, bloating, altered bowel habits, and psychosocial distress, imposing substantial burdens on both individuals and society [3]. Accumulating evidence underscores the pivotal role of the gut microbiota in maintaining gastrointestinal homeostasis and modulating host immune and neural pathways [4,5,6]. In contrast, dysbiosis, which is an imbalance of commensal gut microbiota, has been implicated in the pathophysiology of FBDs [7].

Briefly, the representative dysbiosis are can be summarized as follows: (1) reduced diversity of gut bacteria, (2) an altered ratio of *Firmicutes* to *Bacteroides* (e.g., gut inflammation), and (3) decreased levels of beneficial bacteria (e.g., *Bifidobacterium* and *Lactobacillus*) accompanied by increased levels of harmful bacteria (e.g., gas-producing *Clostridium*) [1]. These perturbations lead to significant dysfunctions, including impaired bile acid metabolism causing diarrhea, altered gut metabolite metabolism causing abdominal bloating, and compromised tight junction integrity resulting in electrolyte loss and subsequent constipation [4,8]. Furthermore, disruption of the gut microbiota can impair the gut–brain axis, a bidirectional communication network involving the central nervous system, neuroendocrine system, immune system, and gut-derived neuroactive metabolites [8,9]. Such impairment may exacerbate visceral hypersensitivity, motility disturbances, and psychosocial symptoms commonly observed in FBDs [10].

Probiotics, defined as live microorganisms that confer health benefits to the host when administered in adequate amounts [11], have received considerable attention as a prevention and treatment agent for gastrointestinal disorders. Evidence from clinical trials indicated that probiotics alleviate FBD symptoms by restoring microbial balance, enhancing intestinal barrier integrity, and modulating immune responses [12,13,14]. However, the heterogeneity in probiotic strains, dosing regimens, and study designs have resulted in inconsistent results, thereby limiting the generalizability of these findings [15]. Moreover, mechanistic studies integrating multi-omics data with clinical outcomes are still limited.

In this study, we aimed to evaluate the clinical efficacy of a multi-strain probiotic complex in adults with FBDs and elucidate its potential mechanisms of action through comprehensive gut microbiota and metabolome profiling.

## 2. Materials and Methods

### 2.1. Ethical Considerations

This clinical trial was conducted at the Inha University Hospital (Incheon, Republic of Korea) between December 2023 and March 2024, in full compliance with the ethical principles outlined in the Declaration of Helsinki and the International Conference on Harmonization Good Clinical Practice (ICH-GCP) guidelines. The study protocol was approved by the Institutional Review Board of Inha University Hospital (Approval No. 2023-10-016) and was registered in the Clinical Research Information Service (CRIS) database (Registration No. KCT0011004), to ensure transparency and ethical oversight throughout the study period. Prior to enrollment, all prospective participants were clearly informed about the objectives of the study, potential therapeutic benefits, possible risks, including unforeseen adverse events, and all relevant study procedures. Written informed consent was obtained from each participant prior to initiation of any study-related procedures to ensure their autonomy and understanding.

### 2.2. Study Participants and Design

Participants were recruited based on strict inclusion and exclusion criteria to ensure the selection of a homogeneous study population representative of individuals suffering from FBDs. Eligible participants were adults aged 19–70 years who fulfilled the Rome IV diagnostic criteria for at least one subtype of FBD, namely functional constipation (FC), functional diarrhea (FD), or functional abdominal bloating and distension (FABD). The exclusion criteria were as follows: comprehensive and included the presence of acute gastrointestinal infections, history of peptic ulcer disease, uncontrolled chronic systemic illnesses, prior gastrointestinal surgeries (excluding appendectomy or cesarean section), and recent use (within 2 weeks) of medications known to influence gastrointestinal motility. Additionally, pregnant or breastfeeding women, individuals with known hypersensitivity to any component of the investigational product, and patients with significant psychiatric conditions, such as major depressive disorder or schizophrenia, were excluded. To avoid potential confounding factors, those with a recent diagnosis (within 2 years) of inflammatory bowel disease, including ulcerative colitis, Crohn’s disease, or colorectal malignancy, were also excluded, as were individuals who had participated in any other clinical trial within the preceding 3 months.

This study employed a randomized, double-blind, placebo-controlled, parallel-group design to minimize bias and ensure the validity of the outcomes. A computer-generated randomization sequence was used to assign participants in a 1:1 ratio to either the probiotic intervention or placebo group. Both groups received capsules that were indistinguishable in appearance, size, color, and packaging to maintain the blinding of participants, clinicians, and data analysts. To further ensure scientific rigor and minimize potential bias, all investigators, participants, and outcome assessors were blinded to group allocation throughout the trial. Each probiotic capsule contained a blend of eight strains, including *Lacticaseibacillus rhamnosus* IDCC 3201 (40%), *Bifidobacterium lactis* IDCC 4301 (15%), *Lacticaseibacillus plantarum* IDCC 3501 (15%), *Limosilactobacillus reuteri* IDCC 3701 (6%), *Bifidobacterium breve* IDCC 4401 (6%), *Lacticaseibacillus casei* IDCC 3451 (6%), *Streptococcus thermophilus* IDCC 2201 (6%), and *Lactobacillus helveticus* IDCC 3801 (6%), with a minimum total viable count of 1 × 10^10^ CFU per capsule. The active blend accounted for part of a capsule (the remainder comprised excipients: corn starch, maltodextrin, silicon dioxide, and magnesium stearate). The participants were instructed to ingest one capsule orally per day for 8 consecutive weeks. Follow-up assessments were scheduled at three time points: baseline (week 0), mid-intervention (week 4), and at the end of the intervention period (week 8). Compliance was carefully monitored by counting the remaining capsules at each visit. Participants who demonstrated an adherence rate of ≥80% were classified as compliant for the per-protocol analysis.

### 2.3. Clinical Assessments

To evaluate the effects of the intervention comprehensively, participants completed validated questionnaires at each study visit. Bowel habits were assessed using a 12-item structured questionnaire that captured data on the weekly frequency of bowel movements, average defecation time, stool quantity and consistency, incidence of irritant bowel movements, feelings of incomplete evacuation, episodes of abdominal pain associated with defecation, severity of abdominal discomfort, flatulence, and postdefecation discomfort. The severity of gastrointestinal symptoms was quantified using the Irritable Bowel Syndrome Severity Scoring System (IBS-SSS), which includes parameters such as the intensity and frequency of abdominal pain, degree of abdominal bloating, dissatisfaction with bowel habits, and impact of symptoms on quality of life. Each parameter was measured using a Visual Analog Scale (VAS) ranging from 0 (no symptoms) to 10 (most severe) to ensure nuanced quantification of symptom severity. Health-related quality of life (QoL) was assessed using a 34-item questionnaire specifically designed to evaluate QoL impairments attributable to bowel dysfunction across the following eight domains: dysphoria, interference with daily activities, body image concerns, health-related anxieties, food avoidance behaviors, social interactions, sexual health, and interpersonal relationships.

### 2.4. Fecal Sample Collection and Storage

Fecal samples were collected at baseline and at weeks 4 and 8 using a standardized AccuStool Collection Kit (AccuGene, Incheon, Republic of Korea). Each kit provided participants with two collection tubes (AccuGene, Incheon, Republic of Korea): one containing a DNA stabilization buffer for microbiota analysis and the other containing an empty sterile tube for metabolomic profiling. Participants were instructed to collect fecal samples within 24 h prior to each study visit and store them at −20 °C until transportation. Upon arrival at the study site, samples were immediately stored at −80 °C for long-term preservation until subsequent analyses.

### 2.5. Microbiome Profiling

For microbiome profiling, genomic DNA was extracted from approximately 250 mg of fecal material using the DNeasy PowerSoil Pro Kit (Qiagen, Hilden, Germany), following the manufacturer’s protocol. DNA concentration and purity were assessed using the Quant-IT PicoGreen dsDNA Assay Kit (Invitrogen, Carlsbad, CA, USA). The full-length 16S rRNA gene (V1–V9 regions) was amplified using barcoded universal primers (27F/1492R) and sequenced using a PacBio Sequel IIe platform (Pacific Biosciences, Menlo Park, CA, USA). Sequencing libraries were prepared with the SMRTbell Express Template Prep Kit 3.0, and high-fidelity reads were generated using the SMRT Link software (v11.0). After quality filtering to remove chimeric and low-quality sequences, taxonomic classification was performed using a curated long-read reference database, enabling species-level resolution. Diversity analyses included alpha diversity indices (Shannon index, Simpson index, and observed species richness) and beta diversity metrics (Bray–Curtis dissimilarity and UniFrac distances), with visualizations generated via principal coordinate analysis and hierarchical clustering using QIIME (v1.9.0) and R software (v4.0.3).

### 2.6. Metabolomic Profiling

For untargeted metabolomic analysis, approximately 20 mg of fecal sample was homogenized in deionized water and methanol (Sigma-Aldrich, St. Louis, MO, USA), followed by centrifugation and filtration. The resulting extracts were derivatized using methoxyamine hydrochloride (Sigma-Aldrich, St. Louis, MO, USA) and N,O-bis (trimethylsilyl)trifluoroacetamide (BSTFA) (Sigma-Aldrich, St. Louis, MO, USA) and analyzed using a Thermo Trace 1310 gas chromatograph (Thermo Fisher Scientific, Waltham, MA, USA) coupled with a Thermo ISQ LT mass spectrometer (Thermo Fisher Scientific, Waltham, MA, USA). Chromatographic separation was achieved using a DB-5MS capillary column (Agilent, Santa Clara, CA, USA) under an optimized temperature gradient. Metabolite identification was performed by matching the spectra against the NIST library and MS-DIAL databases. The data were normalized to the total ion current for subsequent statistical analyses.

### 2.7. Safety Evaluations

Safety monitoring included systematic assessment of adverse events, vital signs (heart rate, respiratory rate, blood pressure, and body temperature), and comprehensive hematological and biochemical laboratory tests. Blood samples were analyzed for complete blood count, liver and renal function markers, electrolytes, and high-sensitivity C-reactive protein to detect inflammatory responses.

### 2.8. Statistical Analysis

All statistical analyses were performed using GraphPad Prism (v10.2.1) and R (v4.0.3). The normality of the data distribution was assessed using the Shapiro–Wilk test. Between-group comparisons of continuous variables were conducted using independent *t*-tests or Mann–Whitney U tests, while within-group changes were evaluated using paired *t*-tests or Wilcoxon signed-rank tests, as appropriate. Categorical variables were analyzed using the chi-square or Fisher’s exact tests. Multivariate analyses of metabolomic data were conducted using MetaboAnalyst 6.0 (https://www.metaboanalyst.ca/ (accessed on 4 March 2025)), and a *p*-value < 0.05 was considered statistically significant.

## 3. Results

### 3.1. Baseline Characteristics

A total of 39 individuals were screened for eligibility, and, after excluding one participant due to withdrawal of consent prior to randomization, 38 participants were successfully enrolled and randomized in a 1:1 ratio into either the probiotic intervention group (n = 19) or the placebo group (n = 19). These participants constituted the full analysis set (FA), as illustrated in the CONSORT flow diagram (Figure 1). During the intervention phase, attrition was observed in both groups, with four participants in the probiotic group and five participants in the placebo group discontinuing treatment for reasons such as intake of prohibited medications, newly confirmed pregnancy, voluntary consent withdrawal, protocol violations, or investigator-determined study termination. Consequently, 15 participants in the probiotic group and 14 in the placebo group completed all study visits and were included in the per-protocol analysis.

Baseline demographic and clinical characteristics, including age, sex, body mass index, smoking and alcohol consumption habits, physical activity levels, sleep duration, and vital signs such as systolic and diastolic blood pressure, were well balanced between the groups, with no statistically significant differences (Table 1). The baseline laboratory parameters did not show any significant differences between the groups.

Treatment adherence, measured as the proportion of capsules consumed relative to the total number provided, was remarkably high in both groups, exceeding 90% at all scheduled visits. None of the participants were excluded because of low compliance (<80%). Importantly, no severe adverse events were reported throughout the study period, and no clinically significant abnormalities were detected in hematological or biochemical safety parameters at any time point in either group.

### 3.2. Questionnaire-Based Outcomes

#### 3.2.1. IBS Symptom Severity (IBS-SSS)

Probiotic intervention significantly improved multiple domains of the IBS-SSS compared to the placebo. At week 8, participants receiving the probiotic complex exhibited a marked reduction in abdominal bloating severity and interference with QoL compared to placebo (*p =* 0.039 and *p =* 0.020, respectively; Figure 2 and Table 2). In within-group analyses, significant decreases were observed in abdominal pain severity (*p =* 0.004), bloating severity (*p =* 0.007), and interference with daily life (*p* = 0.042), highlighting the potential efficacy of the probiotic in alleviating core gastrointestinal symptoms associated with functional bowel disorders (Table 3).

#### 3.2.2. Quality of Life (IBS-QoL)

The probiotic group demonstrated superior improvements in the QoL assessment compared with placebo, particularly in the body image domain at week 8 (*p =* 0.033; Table 4). Within-group comparisons revealed significant enhancements across multiple domains, including dysphoria (*p =* 0.011), interference with activity (*p =* 0.010), interpersonal relations (*p =* 0.025), food avoidance behaviors (*p =* 0.004), social reactions (*p =* 0.009), body image (*p =* 0.002), and health-related concerns (*p =* 0.008), suggesting a comprehensive benefit of the probiotic complex, not only in physical symptoms, but also in psychosocial well-being (Table 5).

#### 3.2.3. Bowel Habit Questionnaire

Although significant improvement in bowel activity in the probiotic group was not observed, compared to the placebo group (Table 6), significant improvements within the probiotic group were observed in bowel function parameters (Table 7). For example, reductions were observed in the frequency of irritant bowel movements (*p =* 0.027), sensation of incomplete evacuation (*p* < 0.001), abdominal pain during defecation (*p* = 0.010), gas production (*p* = 0.012), and postdefecation discomfort (*p =* 0.033).

### 3.3. Gut Microbiome Composition

To assess the effect of probiotic supplementation on gut microbial ecology, we analyzed alpha and beta diversity at baseline, week 4, and week 8 in both the placebo and probiotic groups. Alpha diversity metrics, including Chao1 richness and Shannon and Simpson indices, showed no significant temporal changes within or between groups, indicating that probiotic supplementation did not substantially affect the overall microbial richness or evenness. The preservation of alpha diversity suggests that the intervention exerted targeted effects without disrupting overall ecological complexity at the individual level (Figure 3A). Beta diversity, assessed by Non-metric Multidimensional Scaling based on Bray–Curtis dissimilarity, indicated temporal modulation of the microbial community composition in the probiotic group. While baseline profiles were comparable across groups and no clear separation was observed between treatment arms, samples from weeks 4 and 8 in the probiotic group progressively diverged from the baseline, indicating a time-dependent modulation of community structure. In contrast, microbial communities in the placebo group remained relatively stable (Figure 3B).

Together, these results demonstrate that probiotic administration induced significant compositional remodeling of the gut microbiota over time, reflected in altered beta diversity, while maintaining stable within-sample diversity throughout the intervention period.

Species-level analysis of the gut microbiota composition revealed significant temporal shifts in several bacterial taxa between the placebo and probiotic groups over the 8-week intervention period (Figure 4). Notably, the relative abundance of *Blautia stercoris* was significantly elevated in the probiotic group at week 4 compared to the placebo group (*p* = 0.035), suggesting an early ecological response to supplementation. Similarly, *Faecalibacterium prausnitzii*, a well-established butyrate-producing commensal with anti-inflammatory properties, exhibited a significant increase in abundance at week 8 in the probiotic group versus placebo (*p* = 0.049). These microbial shifts align with the observed clinical improvements and support the potential role of probiotic formulations in enhancing gut microbial resilience and anti-inflammatory capacity.

Conversely, the relative abundance of *Drancourtella massiliensis* was consistently higher in the placebo group at baseline, week 4, and week 8 (*p* = 0.048 and 0.016), suggesting suppression of opportunistic species in response to the probiotic intervention. *Blautia caecimuris* also showed a significantly lower abundance in the probiotic group at week 8, potentially reflecting competitive exclusion dynamics within the gut ecosystem. Furthermore, *Anaerotignum aminivorans* levels were elevated in the placebo group at week 8 (*p* = 0.017), whereas this trend was not observed in the probiotic group.

Collectively, these findings indicate that multi-strain probiotic supplementation leads to selective enrichment of beneficial commensals, such as *F. prausnitzii* and *B. stercoris*, while concurrently suppressing or stabilizing the abundance of less favorable taxa. These microbial alterations may contribute to physiological and psychological benefits in individuals with functional bowel disorders.

### 3.4. Fecal Metabolomic Profiles Differ Significantly Between Placebo and Probiotic Groups

Principal component analysis (PCA) was conducted to assess the global variance in fecal metabolite profiles between the placebo and probiotic groups at baseline and at 4 and 8 weeks. At week 4, partial separation was observed between the two groups along PC1 (15.2%) and PC2 (8.9%), and a similar degree of separation was maintained at week 8 (PC1: 13.5%, PC2: 7.4%). Although overlapping was present, the clustering trend suggested that probiotic intervention contributed to a gradual shift in metabolic composition over time (Figure 5A). PLS-DA was performed to enhance class discrimination. At week 4, the model achieved a clear separation between the placebo and probiotic groups along components 1 (9.9%) and 2 (8.7%). A comparable degree of separation was maintained at week 8, with components 1 and 2 accounting for 8.4% and 8.8% of the variation, respectively (Figure 5B). These results indicate that probiotic supplementation led to significant alterations in fecal metabolite patterns compared to the placebo. Variable Importance in Projection (VIP) analysis from the PLS-DA model identified the most discriminatory metabolites between groups. At week 4, glucose, daidzein, desaminotyrosine and lactic acid exhibited VIP scores > 2.4, indicating strong contributions to group separation. At week 8, metabolites such as lysine (VIP > 4.0), nonanoic acids, and 2-aminobutyric acid were the most influential variables (Figure 5C). These results highlight amino acid metabolism as a key pathway influenced by probiotic intake. Hierarchical clustering heatmaps were used to visualize metabolite abundance across samples. At week 4, the probiotic group exhibited relatively higher levels of short-chain fatty acids compared to placebo. By week 8, more pronounced differences were evident, with metabolites such as L-lysine, nonanoic acid, and serotonin showing differential expression patterns between the groups (Figure 5D). These findings suggest time-dependent modulation of the gut metabolite composition by probiotic treatment. Volcano plots were generated to identify statistically and biologically significant metabolic changes. At week 4, a total of 18 metabolites were significantly altered (log_2_FC > 1 or <−1, *p* < 0.05), including upregulation of L-lactic acid and N-acetyl-L-leucine and downregulation of Gondoic acid in the probiotic group. At week 8, 24 metabolites showed significant changes, with notable increases in L-alanine, 2,4-diaminobutyric acid (DABA), L-norvaline, nonanoic acid, and serotonin, and decreased levels of L-lysine and decanoic acid (Figure 5E and Figure 6).

## 4. Discussion

This double-blind, randomized, placebo-controlled trial demonstrated that an 8-week supplementation of multi-species probiotics significantly alleviated core gastrointestinal symptoms and improved the QoL in individuals with FBDs. These findings were reinforced by microbial and metabolic shifts in the gut environment, underscoring the mechanistic plausibility of this multi-strain probiotic as a gut–brain axis modulator.

The reduction in abdominal pain and bloating observed in the probiotic group is in line with prior reports that multi-strain probiotics outperform mono-strain interventions in ameliorating IBS-related symptoms [12,13,14]. Notably, our within-group comparisons showed statistically significant improvements in IBS-SSS domains, such as abdominal discomfort and interference with daily activities, supporting the therapeutic potential of probiotics in rebalancing the dysbiotic gut ecosystem often implicated in FBD pathophysiology [15]. These improvements mirror findings in other randomized controlled trials, where specific probiotic formulations reduced visceral hypersensitivity and modulated gut motility [16,17].

The observed benefits extend beyond gastrointestinal parameters to include psychosocial domains, as evidenced by significant improvements in QoL metrics such as dysphoria and interpersonal relations. This underscores the involvement of the gut–brain axis, which is increasingly recognized as a critical pathway through which the gut microbiota influences central nervous system processes [18]. Similar outcomes have been reported in meta-analyses, suggesting that specific probiotic strains can act as psychobiotics to improve mood and cognitive performance in patients with IBS [19].

Microbiome profiling revealed enrichment of *Faecalibacterium prausnitzii*, a butyrate-producing commensal bacterium renowned for its anti-inflammatory properties and association with gut barrier integrity [20,21]. Restoration of *F. prausnitzii* is particularly promising because diminished levels of this bacterium have been linked to Crohn’s disease and ulcerative colitis pathogenesis [22]. Additionally, the increased abundance of *Blautia* species, which are known producers of acetate and other SCFAs, indicates a beneficial ecological shift that favors intestinal homeostasis [23]. Although no significant changes were observed in overall diversity metrics, the selective enrichment of beneficial taxa such as *F. prausnitzii* and *B. stercoris* is likely to have contributed to the observed clinical improvements. Consistent with our findings, previous clinical trials have reported that probiotic supplementation frequently induces taxa-specific shifts without altering microbial diversity, and such results have been associated with functional benefits [24,25]. These results suggest that strain- or taxa-specific modulation, rather than community-wide restructuring, may be sufficient to elicit clinically meaningful effects.

Metabolomic analyses further revealed significant elevations in neuroactive compounds, such as serotonin and L-norvaline, which are central to gut–brain axis communication. Approximately 90% of serotonin is produced in the gastrointestinal tract and regulates both enteric motility and central mood states [26]. The increased fecal serotonin concentrations in the probiotic group align with previous findings showing that microbial metabolites can stimulate enterochromaffin cells and enhance serotonergic signaling [27]. L-Norvaline, an arginase inhibitor, has been implicated in the modulation of immune responses and maintenance of intestinal homeostasis in experimental models [28].

Importantly, the increase in DABA observed in this study supports the hypothesis that probiotic intake can modulate gamma-aminobutyric acidergic signaling pathways, potentially attenuating stress-induced visceral pain [29,30]. The interplay between these metabolites and gut microbial taxa highlights the intricate mechanisms by which probiotics exert systemic effects.

In line with these findings, the relative abundance of *Faecalibacterium prausnitzii* has been consistently linked to key human health outcomes, including metabolic syndrome and inflammatory bowel disease. Preclinical evidence indicates that *F. prausnitzii* may attenuate inflammation and normalize mucosal serotonergic (5-HT) signaling [31]. Moreover, prospective cohort studies have demonstrated that higher citrus intake is associated with increased *F. prausnitzii* abundance, accompanied by favorable shifts in monoamine-related markers such as 5-HT and GABA [32]. However, because the present trial did not directly assess metabolite–microbiota relationships, these findings should be considered hypothesis-generating. Future studies with strain-level resolution and mechanistic approaches are required to clarify species- and strain-specific effects. Compared to previous trials, this study uniquely integrated high-resolution microbiome sequencing and untargeted metabolomics to provide mechanistic insights into probiotic actions. Moreover, by assessing psychosocial parameters alongside biological markers, it addresses the multifactorial nature of FBDs more comprehensively than earlier studies [16,19,33].

This study has several limitations that should be considered. First, the relatively small sample size and short intervention period may have limited the statistical power, the generalizability of the findings, and the ability to evaluate long-term durability. Second, because a multi-strain probiotic was used without examining the contribution of each strain separately, and because analyses were based on 16S rRNA sequencing rather than strain-level methods, we were unable to attribute the effects to individual taxa or confirm their persistence. Third, host biomarkers such as cytokines, intestinal permeability, or stress-related indices, as well as objective outcomes including stool transit time or abdominal girth, were not assessed, which constrains mechanistic interpretation and clinical robustness. In addition, the absence of an active comparator arm prevents direct contextualization of probiotic-specific effects, and subgroup analyses by FBD subtype or sex were not feasible given the limited sample size. Lastly, correlations between microbial taxa and metabolite profiles, or ex vivo functional validation, were not performed, limiting causal inference.

Despite these limitations, the present trial provides important evidence that a multi-strain probiotic can improve gastrointestinal symptoms and enhance quality of life in patients with functional bowel disorders, suggesting therapeutic relevance in clinical settings. Beyond symptomatic improvements, the integration of microbiome and metabolomic data revealed consistent shifts in microbial composition and metabolic activity, offering plausible mechanistic explanations for the observed benefits. Taken together, these results provide a valuable foundation for future large-scale and longer-term studies, and highlight the potential of probiotics as microbiota-targeted interventions in functional bowel disorders.

## 5. Conclusions

This randomized, double-blind, placebo-controlled trial demonstrated that 8-week supplementation with a multi-strain probiotic complex and multi-species probiotics significantly alleviated core gastrointestinal symptoms and improved psychosocial well-being in adults with FBDs. The probiotic group showed notable reductions in abdominal pain and bloating as well as enhancements in quality-of-life domains, including emotional and social functioning.

Gut microbiome analysis revealed an increased abundance of beneficial taxa, such as *Faecalibacterium prausnitzii* and *Blautia stercoris*, which are both associated with short-chain fatty acid production and anti-inflammatory effects. Although alpha diversity remained stable, beta diversity indicated temporal shifts in the microbial composition of the probiotic group. Fecal metabolomics further identified elevations in serotonin, L-norvaline, and DABA, suggesting modulation of the gut–brain axis and neuroimmune pathways.

These findings highlight the potential of multispecies probiotics as microbiota-targeted interventions to improve both gastrointestinal and psychological symptoms in FBDs. Future studies with larger cohorts and longer follow-up periods are warranted to validate our results and explore their long-term clinical relevance.

## Figures and Tables

**Figure 1 microorganisms-13-02283-f001:**
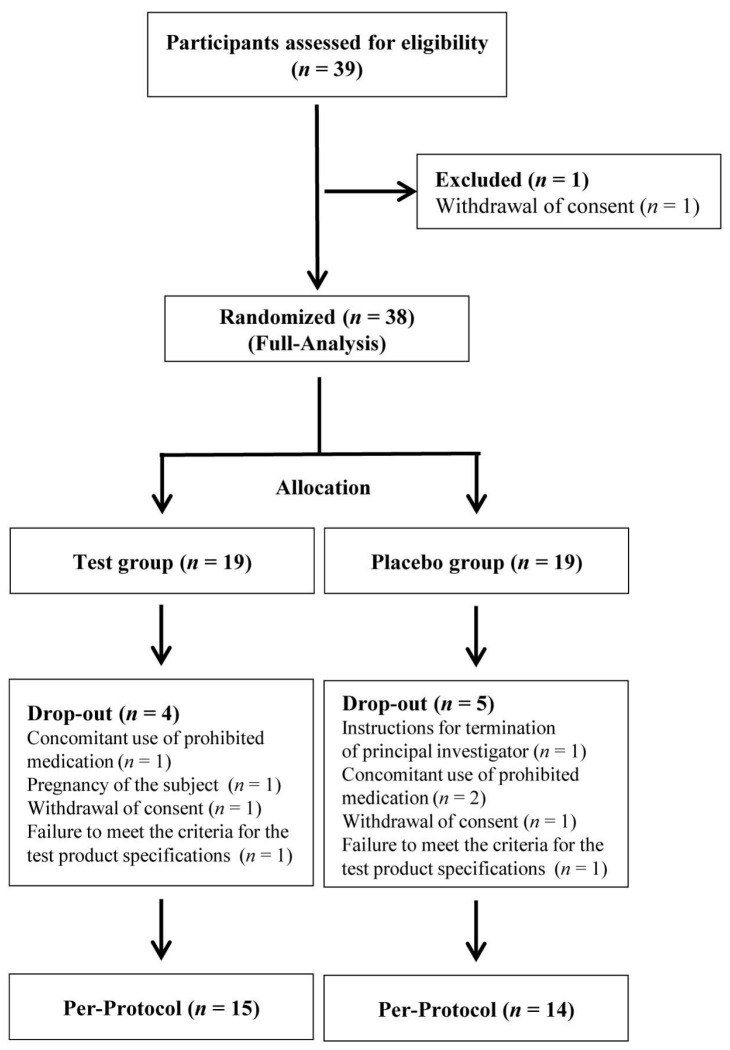
Flow diagram of participant progression through the study. Thirty-nine individuals were screened for eligibility; 1 was excluded due to withdrawal of consent, and 38 participants were randomized into probiotic and placebo groups. During the study, nine participants discontinued for reasons including prohibited medication use and protocol deviations. A total of 15 participants in the probiotic group and 14 in the placebo group completed the trial and were included in the per-protocol analysis. n refers to the sample size.

**Figure 2 microorganisms-13-02283-f002:**
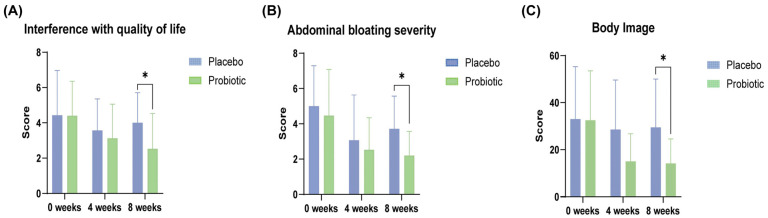
Improvements in gastrointestinal symptoms and quality of life in the probiotic group during the 8-week intervention. (**A**) Interference with quality of life and (**B**) abdominal bloating severity were measured using the IBS-SSS, whereas (**C**) the body image domain score was derived from the IBS-QOL. Data are presented as mean ± SD at baseline, week 4, and week 8. Significant differences between the probiotic and placebo groups are indicated (* *p* < 0.05; Mann–Whitney U test).

**Figure 3 microorganisms-13-02283-f003:**
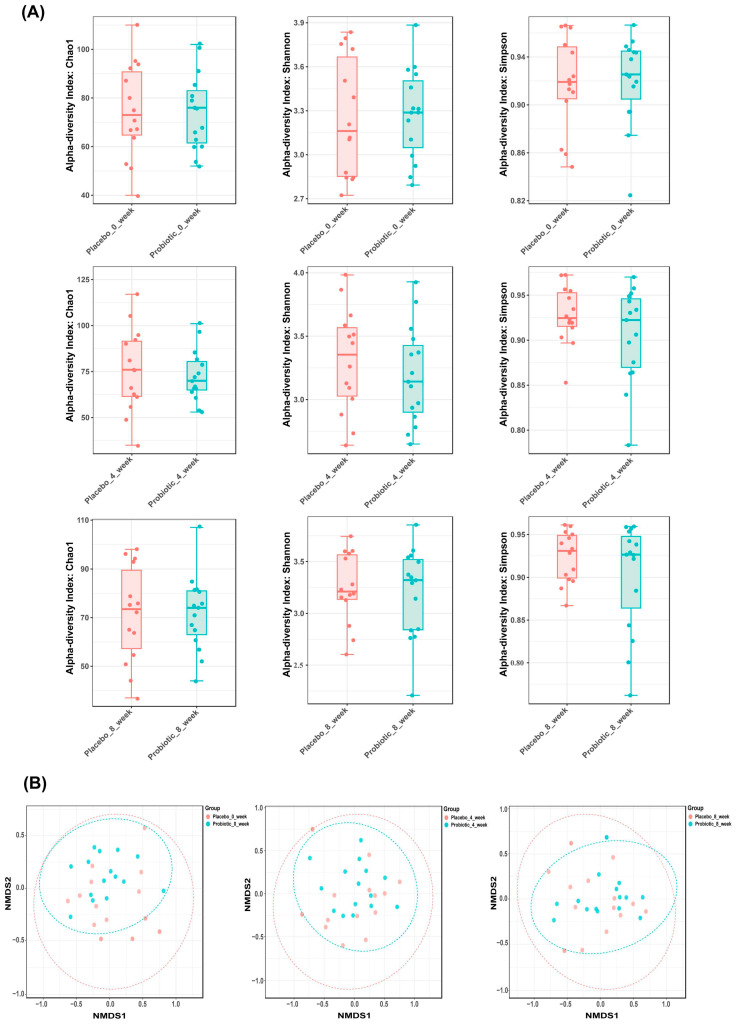
Gut microbiota diversity dynamics in placebo and probiotic groups over time. (**A**) Alpha diversity indices (Chao1, Shannon, Simpson) remained stable across groups. Panels correspond to week 0, 4, and 8 (top to bottom). (**B**) Beta diversity was visualized using Non-metric Multidimensional Scaling plots based on Bray–Curtis dissimilarity. Each point represents the microbial community of an individual participant; closer points indicate higher similarity. Plots correspond to week 0, 4, and 8 (left to right), with temporal clustering observed in the probiotic group, reflecting changes in microbial composition.

**Figure 4 microorganisms-13-02283-f004:**
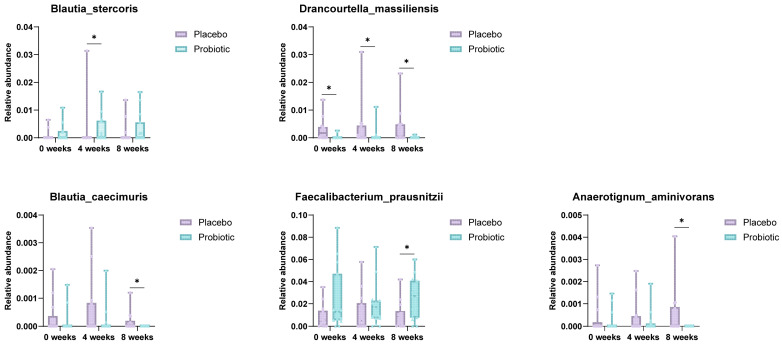
Species-level comparison of gut microbiota between placebo and probiotic groups. Box plots illustrate significantly different species abundances at week 4 and week 8. *Blautia stercoris* and *F. prausnitzii* were notably increased in the probiotic group, suggesting enhanced anti-inflammatory potential (* *p* < 0.05; Mann–Whitney U test).

**Figure 5 microorganisms-13-02283-f005:**
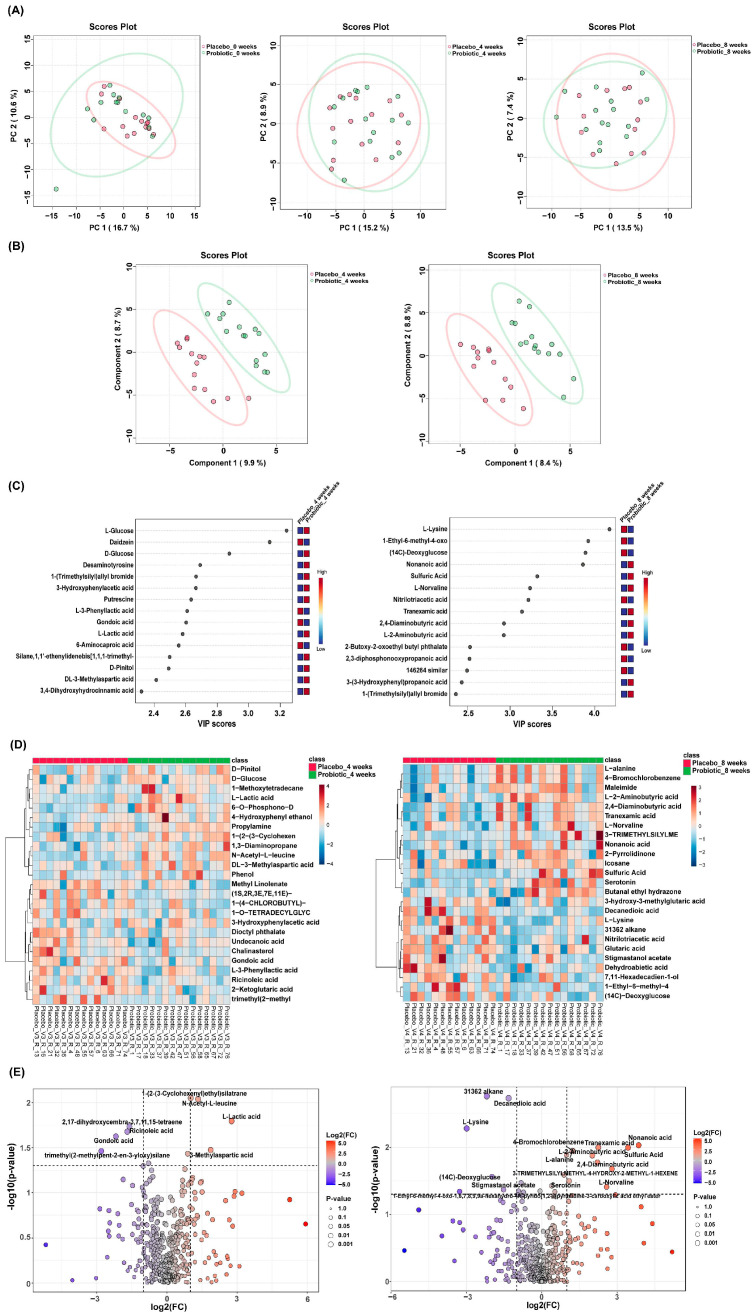
Distinct fecal metabolomic signatures between placebo and probiotic groups. (**A**) PCA and (**B**) PLS-DA showed group separation at weeks 4 and 8. (**C**) VIP plots identified key discriminatory metabolites. (**D**) Heatmaps and (**E**) volcano plots highlighted significantly altered metabolites, with red and blue dots indicating increased and decreased levels in the probiotic group, respectively.

**Figure 6 microorganisms-13-02283-f006:**
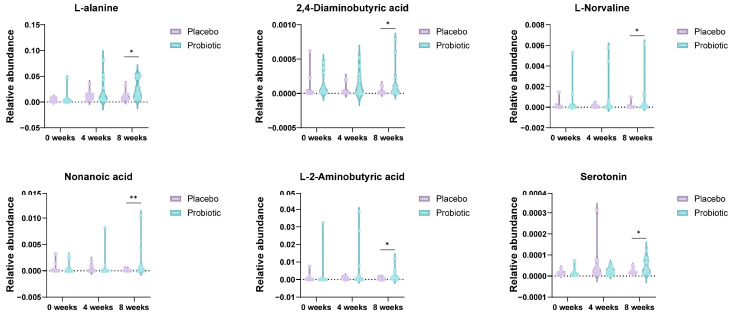
Enrichment of neuroactive and immunomodulatory metabolites in the probiotic group at week 8. Violin plots depict relative abundances of L-alanine, DABA, L-norvaline, nonanoic acid, L(+)-2-aminobutyric acid, and serotonin levels. These metabolites are implicated in gut–brain axis signaling and immune modulation. Significance was determined using the Mann–Whitney U test (* *p* < 0.05; ** *p* < 0.01).

**Table 1 microorganisms-13-02283-t001:** General characteristics of the study groups.

Variables	Placebo Group	Probiotics Group	*p* Values
Sex (n (%))			>0.999
Male	6 (42.9)	6 (40.0)	
Female	8 (57.1)	9 (60.0)	
Age (years)	42.80 ± 13.97	40.88 ± 9.93	0.889
Height (cm)	165.38 ± 6.62	164.94 ± 8.23	0.540
Weight (kg)	69.95 ± 15.78	69.78 ± 17.14	0.583
BMI (kg/m^2^)	25.72 ± 4.70	25.55 ± 5.67	0.923
Smoking (n (%))			>0.999
Yes	0 (0.0)	1 (6.7)	
No	14 (100)	14 (93.3)	
ex	0 (0.0)	0 (0.0)	
Alcohol drinking (n (%))			0.710
Yes	5 (35.7)	7 (46.7)	
No	9 (64.3)	8 (53.3)	
Light-intensity Physical Activity	1.80 ± 1.08	3.00 ± 2.10	0.104
Moderate-intensity Physical Activity	1.33 ± 1.29	2.44 ± 2.16	0.064
Sleep hour	6.57 ± 1.24	6.31 ± 0.87	0.357
Systolic blood pressure (mm Hg)	124.33 ± 13.98	127.06 ± 13.35	0.691
Diastolic blood pressure (mm Hg)	71.93 ± 13.92	75.00 ± 10.77	0.444
Pulse (min)	80.00 ± 10.46	75.88 ± 6.23	0.325
Sex (n (%))			>0.999
Male	6 (42.9)	6 (40.0)	

Values are expressed as means ± standard deviation or n (%). No significant differences were observed between the two groups. *p* values were obtained using the chi-square test or Fisher’s exact test for categorical variables and the Mann–Whitney U test for continuous variables. “ex” indicates individuals who have quit smoking (ex-smokers).

**Table 2 microorganisms-13-02283-t002:** Changes in gastrointestinal symptom severity assessed by the IBS Symptom Severity Scale (IBS-SSS).

	0 Weeks			4 Weeks			8 Weeks		
	Placebo Group	Probiotics Group	*p* Values	Placebo Group	Probiotics Group	*p* Values	Placebo Group	Probiotics Group	*p* Values
Abdominal pain severity	4.64 ± 1.86	4.47 ± 2.17	0.927	3.07 ± 2.13	3.33 ± 2.06	0.718	2.57 ± 1.60	2.07 ± 1.87	0.401
Abdominal pain frequency	2.86 ± 1.70	3.20 ± 1.42	0.582	1.57 ± 1.09	2.00 ± 1.41	0.433	2.00 ± 1.11	1.69 ± 1.29	0.423
Abdominal bloating severity	5.00 ± 2.29	4.47 ± 2.61	0.594	3.07 ± 2.56	2.53 ± 1.81	0.593	**3.71 ± 1.86**	**2.20 ± 1.37**	**0.039 ***
Dissatisfaction of bowel habits	5.14 ± 2.38	5.27 ± 1.28	0.670	4.43 ± 1.40	4.47 ± 1.55	0.887	4.29 ± 1.49	4.00 ± 2.24	0.616
Interference with quality of life	4.43 ± 2.53	4.40 ± 1.96	0.854	3.57 ± 1.79	3.13 ± 1.92	0.638	**4.00 ± 1.71**	**2.53 ± 2.00**	**0.020 ***
Total	22.07 ± 9.39	21.80 ± 6.73	0.957	15.33 ± 5.80	15.47 ± 5.49	0.923	16.57 ± 5.94	12.47 ± 6.06	0.074

Values are expressed as means ± standard deviation or n (%). Significant differences in changes between groups were obtained using the Mann–Whitney U test at * < 0.05.

**Table 3 microorganisms-13-02283-t003:** Within-group changes in gastrointestinal symptom severity assessed by the IBS Symptom Severity Scale (IBS-SSS).

Variables	Probiotics Group								
	0 Weeks	4 Weeks	*p* Values	4 Weeks	8 Weeks	*p* Values	0 Weeks	8 Weeks	*p* Values
Abdominal pain severity	4.47 ± 2.17	3.33 ± 2.06	0.075	3.33 ± 2.06	2.07 ± 1.87	0.060	**4.47 ± 2.17**	**2.07 ± 1.87**	**0.004 ****
Abdominal pain frequency	3.20 ± 1.42	2.00 ± 1.41	0.110	2.00 ± 1.41	1.67 ± 1.29	0.359	**3.20 ± 1.42**	**1.67 ± 1.29**	**0.025 ***
Abdominal bloating severity	**4.47 ± 2.61**	**2.53 ± 1.81**	**0.012 ***	2.53 ± 1.81	2.20 ± 1.37	0.512	**4.47 ± 2.61**	**2.20 ± 1.37**	**0.007 ****
Dissatisfaction of bowel habits	5.27 ± 1.28	4.47 ± 1.55	0.129	4.47 ± 1.55	4.00 ± 2.24	0.305	5.27 ± 1.28	4.00 ± 2.24	0.096
Interference with quality of life	**4.40 ± 1.96**	**3.13 ± 1.92**	**0.046 ***	3.13 ± 1.92	2.53 ± 2.00	0.290	**4.40 ± 1.96**	**2.53 ± 2.00**	**0.042 ***
Total	**21.80 ± 6.73**	**15.47 ± 5.49**	**0.019 ***	**15.47 ± 5.49**	**12.47 ± 6.06**	**0.038 ***	**21.80 ± 6.73**	**12.47 ± 6.06**	**0.005 ****

Values are expressed as means ± standard deviation or n (%). Within-group differences over time were analyzed using the Wilcoxon signed-rank test at * < 0.05, and ** < 0.01.

**Table 4 microorganisms-13-02283-t004:** Changes in quality of life assessed by the IBS Quality of Life (IBS-QOL) questionnaire.

	0 Weeks			4 Weeks			8 Weeks		
	Placebo Group	Probiotics Group	*p* Values	Placebo Group	Probiotics Group	*p* Values	Placebo Group	Probiotics Group	*p* Values
Dysphoria	26.79 ± 20.19	27.05 ± 21.19	>0.999	19.44 ± 19.85	13.53 ± 12.59	0.537	18.26 ± 16.83	11.86 ± 16.44	0.242
Interference with Activity	29.29 ± 24.01	32.33 ± 22.90	0.611	20.00 ± 19.71	15.33 ± 17.16	0.608	19.29 ± 20.27	14.67 ± 19.95	0.589
Interpersonal relations	18.25 ± 16.91	26.11 ± 24.09	0.482	14.29 ± 16.01	9.63 ± 9.67	0.607	12.30 ± 13.68	10.93 ± 12.53	0.921
Food Avoidance	39.87 ± 21.73	35.01 ± 22.75	0.594	28.56 ± 15.92	17.22 ± 20.04	0.071	24.41 ± 17.75	16.67 ± 17.25	0.262
Social Reaction	26.79 ± 19.52	29.17 ± 23.46	0.851	18.75 ± 20.66	19.17 ± 15.57	0.778	18.75 ± 20.66	17.50 ± 16.90	>0.999
Sexual concerns	14.29 ± 18.90	15.00 ± 20.70	0.929	10.71 ± 14.59	7.50 ± 10.35	0.691	8.93 ± 16.57	3.33 ± 5.72	0.700
Body Image	33.04 ± 22.26	32.50 ± 21.02	0.990	28.57 ± 21.05	15.00 ± 11.76	0.078	**29.46 ± 20.57**	**14.17 ± 10.42**	**0.033 ***
Health Worry	39.29 ± 19.52	43.33 ± 24.49	0.444	25.89 ± 20.49	22.50 ± 18.42	0.716	22.32 ± 14.02	22.50 ± 23.24	0.628

Values are expressed as means ± standard deviation or n (%). Significant differences in changes between groups were obtained using the Mann–Whitney U test at * < 0.05.

**Table 5 microorganisms-13-02283-t005:** Within-group changes in quality of life assessed by the IBS Quality of Life (IBS-QOL) questionnaire.

Variables	Probiotics Group								
	0 Weeks	4 Weeks	*p* Values	4 Weeks	8 Weeks	*p* Values	0 Weeks	8 Weeks	*p* Values
Dysphoria	**27.05 ± 21.19**	**13.53 ± 12.59**	**0.010 ***	13.53 ± 12.59	11.86 ± 16.44	0.365	**27.05 ± 21.19**	**11.86 ± 16.44**	**0.011 ***
Interference with Activity	**32.33 ± 22.90**	**15.33 ± 17.16**	**0.002 ****	15.33 ± 17.16	14.67 ± 19.95	0.977	**32.33 ± 22.90**	**14.67 ± 19.95**	**0.010 ***
Interpersonal relations	**26.11 ± 24.09**	**9.63 ± 9.67**	**<0.001 *****	9.63 ± 9.67	10.93 ± 12.53	0.576	**26.11 ± 24.09**	**10.93 ± 12.53**	**0.025 ***
Food Avoidance	**35.01 ± 22.75**	**17.22 ± 20.04**	**0.004 ****	17.22 ± 20.04	16.67 ± 17.25	0.797	**35.01 ± 22.75**	**16.67 ± 17.25**	**0.011 ***
Social Reaction	29.17 ± 23.46	19.17 ± 15.57	0.156	19.17 ± 15.57	17.50 ± 16.90	0.789	29.17 ± 23.46	17.50 ± 16.90	0.090
Sexual concerns	15.00 ± 20.70	7.50 ± 10.35	0.125	7.50 ± 10.35	3.33 ± 5.72	0.250	15.00 ± 20.70	3.33 ± 5.72	0.063
Body Image	**32.50 ± 21.02**	**15.00 ± 11.76**	**0.008 ****	15.00 ± 11.76	14.17 ± 10.42	>0.999	**32.50 ± 21.02**	**14.17 ± 10.42**	**0.004 ****
Health Worry	**43.33 ± 24.49**	**22.50 ± 18.42**	**<0.001 *****	22.50 ± 18.42	22.50 ± 23.24	0.922	**43.33 ± 24.49**	**22.50 ± 23.24**	**0.008 ****

Values are expressed as means ± standard deviation or n (%). Within-group differences over time were analyzed using the Wilcoxon signed-rank test at * < 0.05, ** < 0.01, and *** < 0.001.

**Table 6 microorganisms-13-02283-t006:** Changes in bowel activity assessed by the bowel activity questionnaire.

	0 Weeks			4 Weeks			8 Weeks		
	Placebo Group	Probiotics Group	*p* Values	Placebo Group	Probiotics Group	*p* Values	Placebo Group	Probiotics Group	*p* Values
Number of bowel movement per week	5.93 ± 3.85	6.53 ± 4.10	0.667	6.21 ± 3.33	7.33 ± 3.79	0.366	6.14 ± 3.37	7.00 ± 3.27	0.420
Number of times of irritant bowel movements	4.07 ± 2.40	3.87 ± 2.33	0.802	3.07 ± 1.69	2.93 ± 1.58	0.907	3.21 ± 1.89	2.67 ± 1.68	0.541
Number of times when bowel movements felt incomplete	4.07 ± 2.20	4.53 ± 2.10	0.618	3.29 ± 1.73	3.73 ± 2.34	0.784	3.26 ± 1.95	2.53 ± 1.55	0.328
Number of times of abdominal pains before bowel movements	3.43 ± 2.14	4.40 ± 2.10	0.242	2.43 ± 1.28	3.33 ± 1.99	0.187	2.50 ± 1.09	3.00 ± 2.04	0.645
Number of times of abdominal pains during bowel movements	3.00 ± 1.84	4.33 ± 1.88	0.054	2.00 ± 1.30	3.07 ± 1.83	0.108	2.14 ± 1.10	2.80 ± 2.08	0.462
Degree of abdominal pain	3.50 ± 2.07	4.20 ± 1.78	0.435	2.43 ± 1.16	3.20 ± 1.78	0.282	2.36 ± 1.01	2.93 ± 2.19	0.567
Amount of gas	5.64 ± 2.73	5.80 ± 2.78	0.853	4.86 ± 2.48	4.00 ± 2.62	0.466	3.64 ± 2.44	3.73 ± 2.25	0.887
Discomfort after bowel movements	3.64 ± 2.87	3.67 ± 1.88	0.885	2.64 ± 1.82	3.13 ± 2.39	0.669	2.43 ± 1.99	2.27 ± 1.98	0.921
Discomfort caused by constipation	2.86 ± 2.51	2.53 ± 2.00	0.853	2.50 ± 2.07	2.20 ± 1.82	0.799	2.07 ± 2.09	2.07 ± 2.12	0.930

Values are expressed as means ± standard deviation or n (%). Significant differences in changes between groups were obtained using the Mann–Whitney U test.

**Table 7 microorganisms-13-02283-t007:** Within-group changes in bowel activity assessed by the bowel activity questionnaire.

Variables	Probiotics Group								
	0 Weeks	4 Weeks	*p* Values	4 Weeks	8 Weeks	*p* Values	0 Weeks	8 Weeks	*p* Values
Number of bowel movement per week	6.19 ± 4.20	7.33 ± 3.79	0.313	7.33 ± 3.79	7.00 ± 3.27	0.730	6.19 ± 4.20	7.00 ± 3.27	0.441
Defecation time	2.00 ± 0.73	2.93 ± 1.58	0.453	2.93 ± 1.58	2.67 ± 1.68	0.625	2.00 ± 0.73	2.67 ± 1.68	0.180
Amount of feces	2.56 ± 0.89	3.73 ± 2.34	0.789	3.73 ± 2.34	2.53 ± 1.55	0.188	2.56 ± 0.89	2.53 ± 1.55	0.563
Number of times of irritant bowel movements	4.25 ± 2.72	3.33 ± 1.99	0.132	3.33 ± 1.99	3.00 ± 2.04	0.741	**4.25 ± 2.72**	**3.00 ± 2.04**	**0.027 ***
Number of times when bowel movements felt incomplete	4.88 ± 2.45	3.07 ± 1.83	0.095	**3.07 ± 1.83**	**2.80 ± 2.08**	**0.011 ***	**4.88 ± 2.45**	**2.80 ± 2.08**	**<0.001 *****
Shape of the feces	3.44 ± 1.15	3.20 ± 1.78	>0.999	3.20 ± 1.78	2.93 ± 2.19	>0.999	3.44 ± 1.15	2.93 ± 2.19	0.781
Number of times of abdominal pains before bowel movements	4.56 ± 2.13	4.00 ± 2.62	0.168	4.00 ± 2.62	3.73 ± 2.25	0.343	4.56 ± 2.13	3.73 ± 2.25	0.102
Number of times of abdominal pains during bowel movements	**4.38 ± 1.78**	**3.13 ± 2.39**	**0.016 ***	3.13 ± 2.39	2.27 ± 1.98	0.759	**4.38 ± 1.78**	**2.27 ± 1.98**	**0.010 ***
Degree of abdominal pain	4.31 ± 1.78	2.20 ± 1.82	0.097	2.20 ± 1.82	2.07 ± 2.12	0.344	4.31 ± 1.78	2.07 ± 2.12	0.059
Amount of gas	**6.06 ± 2.89**	**4.38 ± 2.94**	**0.005 ****	4.38 ± 2.94	4.00 ± 2.42	0.716	**6.06 ± 2.89**	**4.00 ± 2.42**	**0.012 ***
Discomfort after bowel movements	4.06 ± 2.41	3.25 ± 2.35	0.281	3.25 ± 2.35	2.31 ± 1.92	0.073	**4.06 ± 2.41**	**2.31 ± 1.92**	**0.033 ***
Discomfort caused by constipation	2.88 ± 2.36	2.50 ± 2.13	0.617	2.50 ± 2.13	2.13 ± 2.06	0.711	2.88 ± 2.36	2.13 ± 2.06	0.516

Values are expressed as means ± standard deviation or n (%). Within-group differences over time were analyzed using the Wilcoxon signed-rank test at * < 0.05, ** < 0.01, and *** < 0.001.

## Data Availability

The original contributions presented in this study are included in the article. Further inquiries can be directed to the corresponding author.

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
