# Peer review of "Therapeutic Modulation of the Gut Microbiome by Supplementation with Probiotics (SCI Microbiome Mix) in Adults with Functional Bowel Disorders: A Randomized, Double-Blind, Placebo-Controlled Trial"

_microorganisms, 2025, doi:10.3390/microorganisms13102283_

Round 1

Reviewer 1 Report

Comments and Suggestions for Authors

This study reports that multi-strain probiotic supplementation alleviates gastrointestinal symptoms and improves psychosocial well-being in patients with FBDs, potentially through remodeling of gut microbiota and modulation of metabolites. However, the mechanistic link between microbial changes (particularly at the species level) and metabolite alterations remains insufficiently explored, and additional correlation analyses are needed to strengthen the conclusions.

  1. From the perspective of β-diversity, no significant changes were observed, nor was there a clear dependency, as the distributions of the two groups could not be separated.
  2. The manuscript presents interesting findings regarding the impact of supplementation on gut microbiota and metabolic features. However, an important aspect that warrants further clarification is the relationship between microbial and metabolic changes. Specifically, it remains unclear whether the observed alterations in the gut microbiota—particularly at the species level—are correlated with the shifts in metabolite profiles. I recommend that the authors perform a correlation analysis to determine these associations, as this would provide a more comprehensive understanding of how supplementation modulates the microbiota to influence metabolic characteristics. To strengthen the analysis, the authors could consider applying multivariate statistical approaches such as canonical correspondence analysis (CCA), redundancy analysis (RDA), or correlation network analysis.

Author Response

Responses to Reviewers’ Comments

Manuscript ID: microorganisms-3866659

Manuscript Title: Therapeutic Modulation of the Gut Microbiome by Supplementation with Probiotics (ATOMY SCI Microbiome mix) in Functional Bowel Disorders: A Randomized, Double-blind, Placebo-controlled Trial

Responses to Reviewer 1’s Comments

We sincerely thank the reviewer for the thoughtful and constructive feedback, which has greatly contributed to the improvement of our manuscript. We have addressed each comment in detail below.

Comment 1. From the perspective of β-diversity, no significant changes were observed, nor was there a clear dependency, as the distributions of the two groups could not be separated.

Response:

We sincerely appreciate this thoughtful comment. In the revised manuscript, we have explicitly clarified that no clear between-group separation was observed in the β-diversity analysis, while highlighting that within-group temporal shifts occurred in the probiotic arm. We carefully refined the wording to ensure accurate and balanced interpretation.
Revision [Lines 306–312]:
“Beta diversity, assessed by Non-metric Multidimensional Scaling based on Bray–Curtis dissimilarity, indicated temporal modulation of the microbial community composition in the probiotic group. While baseline profiles were comparable across groups and no clear separation was observed between treatment arms, samples from weeks 4 and 8 in the probiotic group progressively diverged from the baseline, indicating a time-dependent modulation of community structure. In contrast, microbial communities in the placebo group remained relatively stable (Fig. 3B).”

Comment 2. The manuscript presents interesting findings regarding the impact of supplementation on gut microbiota and metabolic features. However, an important aspect that warrants further clarification is the relationship between microbial and metabolic changes. Specifically, it remains unclear whether the observed alterations in the gut microbiota—particularly at the species level—are correlated with the shifts in metabolite profiles. I recommend that the authors perform a correlation analysis to determine these associations, as this would provide a more comprehensive understanding of how supplementation modulates the microbiota to influence metabolic characteristics. To strengthen the analysis, the authors could consider applying multivariate statistical approaches such as canonical correspondence analysis (CCA), redundancy analysis (RDA), or correlation network analysis.

Response:

We are grateful for this important suggestion. Our analyses demonstrated distinct temporal shifts in both microbial composition and metabolite profiles, including enrichment of Faecalibacterium prausnitzii and Blautia stercoris, as well as increases in serotonin, L-norvaline, and 2,4-diaminobutyric acid. These results indicate that probiotic supplementation influenced both microbiota and its metabolic output. However, direct correlation analyses between specific taxa and metabolites were not performed due to the exploration design and limited sample size of this pilot trial. We fully concur that integrative multivariate approaches would provide stronger mechanistic insight. Accordingly, we have acknowledged this limitation in the Discussion and emphasized that future studies with larger cohorts will incorporate microbiome–metabolome correlation analyses to delineate causal relationships.
Revision [Lines 455–468]:
“This study has several limitations that should be considered. First, the relatively small sample size and short intervention period may have limited the statistical power, the generalizability of the findings, and the ability to evaluate long-term durability. Second, because a multi-strain probiotic was used without examining the contribution of each strain separately, and because analyses were based on 16S rRNA sequencing rather than strain-level methods, we were unable to attribute the effects to individual taxa or confirm their persistence. Third, host biomarkers such as cytokines, intestinal permeability, or stress-related indices, as well as objective outcomes including stool transit time or abdominal girth, were not assessed, which constrains mechanistic interpretation and clinical robustness. In addition, the absence of an active comparator arm prevents direct contextualization of probiotic-specific effects, and subgroup analyses by FBD subtype or sex were not feasible given the limited sample size. Lastly, correlations between microbial taxa and metabolite profiles, or ex vivo functional validation, were not performed, limiting causal inference.”

Reviewer 2 Report

Comments and Suggestions for Authors

Dear Editor,

I have carefully reviewed the manuscript submitted by Bang et al., which investigates the effects of a multi-strain probiotic (ATOMY SCI Microbiome mix) on clinical and microbiome outcomes in patients with functional bowel disorders (FBDs). The study employs a randomized, double-blind, placebo-controlled trial design and integrates clinical symptom scoring, microbiome sequencing, and metabolomic profiling.

1) Overall Assessment

This manuscript presents a well-designed and timely study that addresses a highly relevant clinical problem. The integration of microbiota and metabolomic analyses strengthens the mechanistic interpretation of the results. The findings are promising and suggest a potential role of multi-strain probiotics in alleviating FBD symptoms and improving psychosocial well-being.

2) Major Comments
While the increase in Faecalibacterium prausnitzii and Blautia stercoris is compelling, the absence of significant alpha diversity changes warrants further contextualization. Could this indicate strain-specific rather than community-wide effects?
Given that the study is partially supported by Ildong Bioscience, the authors should elaborate on measures taken to avoid bias in data collection and interpretation.

3) Figures could be more self-explanatory; for instance, axis labels and legends in the microbiome diversity plots could be clarified for non-specialist readers.

Please standardize reporting of p-values (e.g., “p = 0.039” vs. “p < 0.05”).

Author Response

Responses to Reviewer 2’s Comments

We sincerely appreciate the reviewer’s valuable and insightful comments, which have been highly helpful in refining and strengthening our manuscript. We have carefully considered each point raised and provided detailed responses and corresponding revisions below.

Comment 1. Overall Assessment

This manuscript presents a well-designed and timely study that addresses a highly relevant clinical problem. The integration of microbiota and metabolomic analyses strengthens the mechanistic interpretation of the results. The findings are promising and suggest a potential role of multi-strain probiotics in alleviating FBD symptoms and improving psychosocial well-being.

Response:

We are deeply appreciative of the reviewer’s positive evaluation of our study design, methodological rigor, and clinical relevance. Your recognition of the value of integrating microbiota and metabolomic analyses has been highly motivating and has guided our refinement of the manuscript.

This acknowledgment has been reflected in the revised text.

Comment 2. Major Comments 1

While the increase in Faecalibacterium prausnitzii and Blautia stercoris is compelling, the absence of significant alpha diversity changes warrants further contextualization. Could this indicate strain-specific rather than community-wide effects?

Response:

We thank the reviewer for this perceptive observation. We agree that the absence of significant alpha diversity changes likely reflects selective, strain-specific modulation rather than broad community-wide restructuring. In the revised manuscript, we emphasized that enrichment of beneficial taxa such as F. prausnitzii and B. stercoris may underlie the observed clinical benefits, even without shifts in overall diversity. Supporting evidence from previous clinical trials reporting similar taxa-specific effects has been added.
Revision [Lines 429–436]:
“Although no significant changes were observed in overall diversity metrics, the selective enrichment of beneficial taxa such as F. prausnitzii and B. stercoris is likely to have contributed to the observed clinical improvements. Consistent with our findings, previous clinical trials have reported that probiotic supplementation frequently induces taxa-specific shifts without altering microbial diversity, and such results have been associated with functional benefits [24, 25]. These results suggest that strain- or taxa-specific modulation, rather than community-wide restructuring, may be sufficient to elicit clinically meaningful effects.”

Comment 3. Major Comments 2

Given that the study is partially supported by Ildong Bioscience, the authors should elaborate on measures taken to avoid bias in data collection and interpretation.

Response:

We greatly appreciate the reviewer’s attention to this important issue. To ensure scientific rigor and transparency, the trial was conducted under strict double-blind conditions with rigorous randomization procedures. Microbiome and metabolomic analyses were performed independently by blinded laboratory personnel, and statistical analyses were conducted by an external biostatistician unaffiliated with Ildong Bioscience. We have also completed and submitted the journal’s conflict-of-interest disclosure forms to ensure full transparency.
Revision [Lines 104–106]:
“To further ensure scientific rigor and minimize potential bias, all investigators, participants, and outcome assessors were blinded to group allocation throughout the trial.”

Comment 4. Major Comments 3

Figures could be more self-explanatory; for instance, axis labels and legends in the microbiome diversity plots could be clarified for non-specialist readers.

Response:

We thank the reviewer for this valuable recommendation. Figures have been revised for improved clarity. Axis labels are now more descriptive, and legends and captions have been expanded with detailed explanations, making them more accessible for readers unfamiliar with microbiome research.
Revision [Figure 3, Lines 313–319]:
“Gut microbiota diversity dynamics in placebo and probiotic groups over time. (A) Alpha diversity indices (Chao1, Shannon, Simpson) remained stable across groups. Panels correspond to week 0, 4, and 8 (top to bottom). (B) Beta diversity was visualized using Non-metric Multidimensional Scaling plots based on Bray–Curtis dissimilarity. Each point represents the microbial community of an individual participant; closer points indicate higher similarity. Plots correspond to week 0, 4, and 8 (left to right), with temporal clustering observed in the probiotic group, reflecting changes in microbial composition.”

Comment 5. Major Comments 4

Please standardize reporting of p-values (e.g., “p = 0.039” vs. “p < 0.05”).

Response:

We are grateful for this careful observation. All p-values throughout the manuscript have been standardized to the uniform format “p = X.XXX” for clarity and consistency.
These revisions have been applied throughout the manuscript.

Reviewer 3 Report

Comments and Suggestions for Authors

The article reports on a randomized, double-blind, placebo-controlled clinical trial investigating the effects of an 8-week supplementation with a multi-strain probiotic mix (ATOMY SCI Microbiome mix) in adults with functional bowel disorders (FBDs). Clinical outcomes: Probiotics reduced abdominal bloating, abdominal pain, and improved quality of life (especially body image, dysphoria, interpersonal relations, and food avoidance). Bowel habits: Improvements were seen within the probiotic group (fewer incomplete evacuations, less abdominal pain during defecation, reduced gas, less post-defecation discomfort). Gut microbiota: Increased relative abundance of Faecalibacterium prausnitzii (week 8) and Blautia stercoris (week 4). These taxa are linked to short-chain fatty acid production and anti-inflammatory effects. Metabolomics: Significant changes in fecal metabolites, including increases in serotonin, L-norvaline, DABA, and other compounds relevant to the gut–brain axis and immune modulation. Safety: No severe adverse events; adherence >90%. Conclusion: Multi-strain probiotics improved gastrointestinal symptoms and psychosocial well-being in FBD patients, likely through gut microbiota remodeling and metabolite modulation (gut–brain axis effects).

Overall: The study provides strong preliminary evidence that multi-strain probiotics can improve both gut and psychosocial symptoms in FBDs, but larger, longer, and more mechanistic studies are needed for clinical translation.

 Suggestions for Improvement

Sample Size & Power = Only 38 participants were enrolled, with 29 completing per protocol. Larger, more diverse cohorts would strengthen statistical power and generalizability.

So- Increase Statistical Power

  • Enroll at least 100+ participants to reduce type II errors.
  • Stratify randomization by FBD subtype (IBS-C, IBS-D, bloating-predominant, etc.) to ensure balanced groups.

Long-term Follow-up =Effects were measured only during 8 weeks. Assessing persistence of benefits over months would clarify durability.

You should - Extend Follow-up

  • Add 12–24 week follow-up visits after discontinuing probiotics to check if microbiota and metabolite changes persist or regress.
  • Assess whether participants relapse once supplementation stops.

Strain-specific Contributions = The probiotic mix contained 8 strains, but the individual role of each was not disentangled. Future studies could test subsets or single strains.

I believe, disentangle Strain-Specific Effects may be added

  • Test single-strain arms (e.g., L. rhamnosus only, B. lactis only) alongside the multi-strain mix.
  • Use shotgun metagenomics instead of just 16S rRNA to confirm colonization and strain-level persistence.

Mechanistic Depth= While metabolomics and microbiota profiling were included, integration with host biomarkers (immune markers, intestinal permeability tests) would provide stronger mechanistic insight.

So- Integrate Host Biomarkers that may be added

  • Measure serum cytokines (IL-6, TNF-α, IL-10) and fecal calprotectin for inflammation.
  • Assess intestinal permeability (lactulose/mannitol test) to link microbiota shifts with gut barrier function.
  • Collect cortisol or HRV (heart rate variability) to connect gut–brain axis changes with stress physiology.

Broader Clinical Endpoints= Objective measures (e.g., stool transit time, imaging, inflammatory markers) could complement subjective questionnaires (IBS-SSS, IBS-QoL).

Strengthen Clinical Assessments

  • Add objective endpoints:
    • Stool consistency (Bristol stool scale with digital photos).
    • Transit time via wireless motility capsule.
    • Abdominal distension via girth measurement.
  • Combine with patient-reported outcomes for robustness.

Comparative Interventions=Including an active comparator (e.g., dietary fiber supplement or another probiotic formulation) could contextualize efficacy better than placebo alone.

Compare Against Active Control

  • Add a fiber/prebiotic arm (e.g., inulin or resistant starch) to distinguish probiotic-specific vs. general microbiota-supportive effects.

Personalized Responses=Subgroup analyses (e.g., constipation- vs diarrhea-predominant FBDs) might uncover differential benefits.

Personalized Analyses

  • Perform responder vs. non-responder analysis to identify baseline microbiota signatures that predict benefit.
  • Explore sex-specific differences since IBS prevalence and microbiota composition vary by sex.

Mechanistic Validation

  • Correlate Faecalibacterium prausnitzii and Blautia stercoris abundance directly with butyrate/acetate levels.
  • Conduct ex vivo assays (e.g., fecal water on epithelial cell cultures) to test anti-inflammatory effects.

Author Response

Responses to Reviewer 3’s Comments

We sincerely appreciate the reviewer’s valuable and insightful comments, which have been highly helpful in refining and strengthening our manuscript. We have carefully considered each point raised and provided detailed responses and corresponding revisions below.

Comment 1. Sample Size & Power = Only 38 participants were enrolled, with 29 completing per protocol. Larger, more diverse cohorts would strengthen statistical power and generalizability. So- Increase Statistical Power, Enroll at least 100+ participants to reduce type II errors. Stratify randomization by FBD subtype (IBS-C, IBS-D, bloating-predominant, etc.) to ensure balanced groups.

Response:

We sincerely appreciate this valuable comment. We fully acknowledge that the relatively small sample size may restrict statistical power and generalizability. Nevertheless, despite the modest cohort, our study detected statistically significant improvements in abdominal bloating (p = 0.039) and QoL interference (p = 0.020), indicating that the trial was adequately powered to capture clinically meaningful effects.
Revision [Lines 455–457]:
“This study has several limitations that should be considered. First, the relatively small sample size and short intervention period may have limited the statistical power, the generalizability of the findings, and the ability to evaluate long-term durability.”

Comment 2. Long-term Follow-up =Effects were measured only during 8 weeks. Assessing persistence of benefits over months would clarify durability. You should - Extend Follow-up. Add 12–24 week follow-up visits after discontinuing probiotics to check if microbiota and metabolite changes persist or regress. Assess whether participants relapse once supplementation stops.

Response:

Response:
We appreciate this thoughtful suggestion. While the trial duration was limited to 8 weeks, both clinical outcomes and microbiome–metabolome profiles demonstrated consistent temporal trajectories, with progressive improvements observed at weeks 4 and 8. These findings suggest sustained effects within the study period. We agree, however, that long-term durability cannot be assessed from this dataset and have acknowledged this limitation.
Revision [Lines 455–457]:
“This study has several limitations that should be considered. First, the relatively small sample size and short intervention period may have limited the statistical power, the generalizability of the findings, and the ability to evaluate long-term durability.”

Comment 3. Strain-specific Contributions = The probiotic mix contained 8 strains, but the individual role of each was not disentangled. Future studies could test subsets or single strains.

I believe, disentangle Strain-Specific Effects may be added, Test single-strain arms (e.g., L. rhamnosus only, B. lactis only) alongside the multi-strain mix. Use shotgun metagenomics instead of just 16S rRNA to confirm colonization and strain-level persistence.

Response:

We thank the reviewer for this insightful comment. While strain-specific effects were not disentangled in this study, species-level analyses revealed selective enrichment of F. prausnitzii (week 8) and B. stercoris (week 4) in the probiotic group. These taxa are well recognized for their beneficial functions and may have contributed to the observed clinical improvements, suggesting potential strain-level contributions.
Revision [Lines 457–461]:
“Second, because a multi-strain probiotic was used without examining the contribution of each strain separately, and because analyses were based on 16S rRNA sequencing rather than strain-level methods, we were unable to attribute the effects to individual taxa or confirm their persistence.”

Comment 4. Mechanistic Depth= While metabolomics and microbiota profiling were included, integration with host biomarkers (immune markers, intestinal permeability tests) would provide stronger mechanistic insight.

So- Integrate Host Biomarkers that may be added, Measure serum cytokines (IL-6, TNF-α, IL-10) and fecal calprotectin for inflammation. Assess intestinal permeability (lactulose/mannitol test) to link microbiota shifts with gut barrier function. Collect cortisol or HRV (heart rate variability) to connect gut–brain axis changes with stress physiology.

Response:

We are grateful for this valuable recommendation. Although host biomarkers such as cytokines or intestinal permeability tests were not measured, our fecal metabolomics analysis revealed increased levels of serotonin, L-norvaline, and 2,4-diaminobutyric acid, metabolites strongly associated with gut–brain and immune signaling. These findings provide partial mechanistic plausibility for the observed effects.
Revision [Lines 461–464]:
“Third, host biomarkers such as cytokines, intestinal permeability, or stress-related indices, as well as objective outcomes including stool transit time or abdominal girth, were not assessed, which constrains mechanistic interpretation and clinical robustness.”

Comment 5. Broader Clinical Endpoints= Objective measures (e.g., stool transit time, imaging, inflammatory markers) could complement subjective questionnaires (IBS-SSS, IBS-QoL).

Strengthen Clinical Assessments, Add objective endpoints: Stool consistency (Bristol stool scale with digital photos). Transit time via wireless motility capsule. Abdominal distension via girth measurement. Combine with patient-reported outcomes for robustness.

Response:

We appreciate this important comment. Although objective measures such as imaging or motility capsules were not included, our bowel habit questionnaire captured functional endpoints beyond subjective perception, showing significant improvements in incomplete evacuation, defecation pain, bloating, and gas-related discomfort (all p < 0.05). These outcomes provide supportive evidence of functional improvement.
Revision [Lines 461–464]:
“Third, host biomarkers such as cytokines, intestinal permeability, or stress-related indices, as well as objective outcomes including stool transit time or abdominal girth, were not assessed, which constrains mechanistic interpretation and clinical robustness.”

Comment 6. Comparative Interventions=Including an active comparator (e.g., dietary fiber supplement or another probiotic formulation) could contextualize efficacy better than placebo alone.

Compare Against Active Control, Add a fiber/prebiotic arm (e.g., inulin or resistant starch) to distinguish probiotic-specific vs. general microbiota-supportive effects.

Response:

We thank the reviewer for raising this valuable point. While this study was designed as a placebo-controlled feasibility trial, significant differences between probiotic and placebo groups across multiple endpoints strongly suggest a probiotic-specific effect. We have acknowledged the absence of an active comparator as a limitation.
Revision [Lines 464–466]:
“In addition, the absence of an active comparator arm prevents direct contextualization of probiotic-specific effects, and subgroup analyses by FBD subtype or sex were not feasible given the limited sample size.”

Comment 7. Personalized Responses=Subgroup analyses (e.g., constipation- vs diarrhea-predominant FBDs) might uncover differential benefits.

Personalized Analyses, Perform responder vs. non-responder analysis to identify baseline microbiota signatures that predict benefit. Explore sex-specific differences since IBS prevalence and microbiota composition vary by sex.

Response:

We greatly appreciate this important suggestion. While subgroup analyses were not feasible due to the limited sample size, our beta diversity analysis showed heterogeneous response trajectories, with progressive divergence from baseline in the probiotic group. This pattern suggests the potential for personalized effects, which should be explored in future larger-scale trials.
Revision [Lines 464–468]:
“In addition, the absence of an active comparator arm prevents direct contextualization of probiotic-specific effects, and subgroup analyses by FBD subtype or sex were not feasible given the limited sample size. Lastly, correlations between microbial taxa and metabolite profiles, or ex vivo functional validation, were not performed, limiting causal inference.”

Comment 8. Mechanistic Validation.

Correlate Faecalibacterium prausnitzii and Blautia stercoris abundance directly with butyrate/acetate levels. Conduct ex vivo assays (e.g., fecal water on epithelial cell cultures) to test anti-inflammatory effects.

Response:

We thank the reviewer for this valuable suggestion. We fully agree that establishing mechanistic links between microbial changes and host-relevant outcomes is essential. While direct correlations between F. prausnitzii/B. stercoris and butyrate/acetate were not performed, nor ex vivo assays, our fecal metabolomics analysis revealed increased levels of serotonin, L-norvaline, and 2,4-diaminobutyric acid—metabolites relevant to gut–brain and immune regulation. These findings provide partial mechanistic plausibility despite the absence of direct correlation testing.

We have acknowledged this limitation in the Discussion and clarified that future studies will integrate microbiome and metabolomics datasets to directly assess correlations between SCFA concentrations and specific taxa, and to perform ex vivo assays to evaluate the anti-inflammatory effects of metabolite shifts.

Revision [Lines 466–468]:
“Lastly, correlations between microbial taxa and metabolite profiles, or ex vivo functional validation, were not performed, limiting causal inference.”

Round 2

Reviewer 3 Report

Comments and Suggestions for Authors

I am satisfied with the modificatiins.